# Detection of Cyclomodulin CNF-1 Toxin-Producing Strains of *Escherichia coli* in Pig Kidneys at a Slaughterhouse

**DOI:** 10.3390/microorganisms11082065

**Published:** 2023-08-11

**Authors:** Arturo Herrera-Vázquez, Rebeca Arellano-Aranda, Daniel Hernández-Cueto, Esmeralda Rodríguez-Miranda, Sergio López-Briones, Marco Antonio Hernández-Luna

**Affiliations:** 1Department of Medicine and Nutrition, Division of Health Sciences, University of Guanajuato, Campus León, Guanajuato 37670, Mexicoerodriguez@ugto.mx (E.R.-M.); lobrisug@ugto.mx (S.L.-B.); 2Department of Veterinary, Division of Life Sciences, University of Guanajuato, Campus Irapuato Salamanca, Guanajuato 36500, Mexico; 3Unit of Investigative Research on Oncological Diseases, Children’s Hospital of Mexico Federico Gomez, Mexico City 06720, Mexico

**Keywords:** CNF-1, *Escherichia coli*, pig kidney, urinary tract infection

## Abstract

Food is often contaminated with *Escherichia coli (E. coli)* bacteria strains, which have been associated with different diseases, including urinary tract infections. The consumption of meat by humans is a potential route of transmission of antimicrobial resistance, and food-producing animals have been associated as a major reservoir of resistant bacterial strains. The aim of this study was to determine the presence of the *E. coli* strains producing the CNF-1 toxin in pig kidneys. Pig kidneys were collected from a Mexican slaughterhouse and classified according to their coloration into reddish kidneys (RK) and yellowish kidneys (YK). A tissue sample from each kidney was processed for histological analysis, the presence of *E. coli* was determined by conventional PCR assay, and the CNF-1 toxin was detected by both conventional PCR and Western blotting. Herein, an inflammatory cell infiltrate was found in all collected kidneys, regardless of macroscopic differences. Surprisingly, *E. coli* and the CNF-1 toxin were detected in all kidney samples. We clearly demonstrate contamination by CNF-1 toxin-producing *E. coli* in pork kidneys from a slaughterhouse, even in those without apparent damage. This suggests that pork may serve as a reservoir for pathogens, representing an important risk to human health.

## 1. Introduction

Foodborne illness represents a risk to human health and an important worldwide public health problem [1]. According to the World Health Organization (WHO), more than 400,000 people die annually from foodborne illness [2].

In the slaughtering procedure, as well as in food processing, contamination with pathogenic microorganisms such as *Salmonella* spp., *Escherichia coli*, *Campylobacter* spp. and other bacteria can occur [3]; this is of widespread concern because it can be a major cause of foodborne illness, directly affecting the health of consumers. Thus, the food industry faces an important challenge, employing different decontamination methods during the slaughter, as well as during the production process of meat products, with the intention of reducing the risk of contamination by pathogenic microorganisms and improving both the quality and safety of meat products [4]. However, limitations can still be found in decontamination methods, which can alter the organoleptic properties of treated products [5,6,7]. In addition, practices can pose health and safety risks for handlers, not only due to the physical and chemical agents used in these processes, but also through exposure to treatment-resistant bacteria, such as *E. coli* O157:H7, an acid-resistant bacterial strain [8].

Commonly, it has been found that food is often contaminated with *E. coli* bacteria strains, and bacterial resistance to antibiotics is a major problem associated with illnesses caused by ingestion of contaminated food [9,10,11,12]. Thus, it has been identified that among the most important foodborne bacterial strains are *E. coli* spp. and the Shiga toxin-producing *E. coli* strain or STEC, both associated with gastroenteritis in humans after ingestion of contaminated pork [13,14].

On the other hand, high genetic homology between STEC and the uropathogenic *E. coli* (UPEC) strain has been found, and UPEC is a bacterium widely associated with urinary tract infections [9,15]. In addition, a higher incidence of urinary tract infections caused by antimicrobial-resistant *E. coli* has been found in women who frequently consume chicken and pork [16], suggesting a possible transmission of UPEC through consumption of contaminated food [17]. Interestingly, antibiotic resistance of *E. coli* strains in samples isolated from a pig farm was mainly related to the presence of plasmids [18]. However, other virulence factors such as hemolysin, mannose-resistant hemagglutinin (MRHA) and mannose-sensitive hemagglutinin (MSHA) have also been associated with antibiotic resistance in humans [19] and swine [20]. Also, UPEC and other pathogenic strains of *E. coli* can produce an important virulence factor, a cyclomodulin called CNF-1 (Cytotoxic Necrotizing Factor-1), which has been associated with increased bacterial survivability [21,22], meningitis [23] and, recently, prostate cancer and metastasis [24,25].

Because of the human health implications of pathogenic *E. coli* strain contamination, the objective of this study was to determine the possible presence of CNF-1 toxin-producing *E. coli* in pork kidney samples from a slaughterhouse, given the common use of viscera in the preparation and production of human food, such as sausages.

## 2. Materials and Methods

### 2.1. Pig Kidney Samples

According to the staff at the slaughterhouse in Irapuato, México, all the pigs arrived at the slaughterhouse apparently healthy, without signs of any disease. However, in the process of butchering, it was found that several pigs had yellowish kidneys (with visible damage like hydronephrosis), while other pigs had normal reddish kidney coloration. Thus, 24 samples of pig kidneys were collected and classified according to their coloration; twelve reddish kidneys (RK) and twelve yellowish kidneys (YK) were obtained. All kidney samples were randomly collected by slaughterhouse personnel according to the Mexican official standard NOM 194 SSA1 2004. For biosafety, the kidneys were individually wrapped in aluminum foil. Each kidney was individually placed in an airtight plastic bag and immediately transported to the laboratory on ice into a sealed cooler. Samples were processed immediately or in no more than 90 min. A tissue sample from each kidney was processed for histological analysis, total DNA extraction and total protein extraction as described below.

### 2.2. Kidney Histology

The pig kidneys were fixed in formaldehyde and dehydrated by xylene and ethanol treatment. Kidney tissues were embedded in paraffin and 3 µm sections were cut from fixed embedded tissues on a rotary microtome (Leica Byosistems™, Wetzlar, Germany). Tissue slices were placed on glass slides, deparaffinized and stained with hematoxylin and eosin (H and E). For histological examination, cellular infiltration was analyzed using conventional light microscopy at 20× and 40× magnification (Leica Byosistems™, Wetzlar, Germany). Three observers counted 10 different fields from each kidney sample. Data were expressed as mean ± SE, and statistical differences between reddish kidneys and yellowish kidneys were determined by Mann–Whitney U.

### 2.3. Kidney DNA Purification

Total DNA was purified from 100 mg of tissue obtained from the renal cortex. Briefly, the tissues were homogenized in 500 µL of TRIzol, (Invitrogen™, Waltham, MA, USA) through sonication for 5 cycles of 30 s at 4 °C, and DNA was purified according to the manufacturer’s protocol.

### 2.4. Detection of Bacterial Genes by PCR

The detection of bacterial genes in pig kidney tissues was carried out by conventional PCR in a thermal cycler T100 (Bio-Rad™, Hercules, CA, USA). The PCR products were subjected to electrophoresis on 1% agarose gels with in TAE buffer (40 mM Tris-acetate and 1 mM EDTA, pH 8.3), stained with SYBR Safe DNA Gel Stain (Invitrogen™, Waltham, MA, USA) and visualized on a SmartDoc Gel Imaging System (Accuris Instruments, Edison, NJ, USA) under blue light. The specific oligonucleotide primers used in PCR amplification were as follows: *E. coli*: Fwd 5′ TTGCTGACGAGTGGCGGACG 3′ and Rv 5′ CCCCACTGCTGCCTCCCGTA 3′ and *cnf-1* gene, Fwd 5′ AGATGGAGTTTCCTATGCAGGAG 3′ and Rv 5′ CATTCAGAGTCCTGCCCTCATTATT 3′ [26]. The universal primers Fwd 5′ TCCTACGGGAGGCAGCAGT 3′ and Rv 5′ GGACTACCAGGGTATCTAATCCTGTT 3′ for the 16S ribosomal subunit, were used as an internal control.

### 2.5. Kidney Protein Purification

Total kidney proteins were obtained from 100 mg of renal cortex tissue. Tissues were homogenized in 500 μL of cold PBS in a Dounce tissue grinder with a glass pestle. The samples were then centrifuged at 500× *g* for 10 min at 4 °C and the supernatant was removed. The pellet was suspended in 1 mL of whole cell lysis buffer containing: 1 M KCl, 1 M HEPES (pH 7.8), 1% NP-40 and a halt protease and phosphatase inhibitor cocktail (Pierce™, Appleton, WI, USA) and sonicated at 4 °C. After that, samples were centrifuged at 9000× *g* for 10 min at 4 °C. Protein concentration in the supernatants was quantified by Bradford assay. All protein samples were stored at −70 °C until use.

### 2.6. Western Immunobloting Assay

Total kidney proteins (40 µg) were separated by 10% SDS-PAGE and transferred to a PDVF membrane in a TransBlot turbo (Bio-Rad™, Hercules, CA, USA) following the manufacturer’s protocols. The PDVF membranes were incubated 2 h at room temperature in phosphate buffered saline containing 0.05% Tween-20 and 5% non-fat dry milk (PBS-T) to block nonspecific antibody binding. The membranes were then probed with primary either anti-CNF 1 (abcam™, Cambridge, CB2 0AX, UK) or anti-β actin (Santa Cruz, CA, USA, Biotechnology™) antibodies (1:3000) overnight at 4 °C. After four washes with PBS-T, complexes were detected by incubation for 1 h at room temperature with the appropriate secondary antibody using an enhanced chemiluminescence protein detection kit, Wester Lightning Plus-ECL (Perkin Elmer™, Waltham, MA, USA).

### 2.7. Ethical Approval Statement

The protocol was approved by the institutional bioethics committee of the University of Guanajuato. All experiments and procedures were performed in accordance with current Mexican legislations (NOM-062-ZOO-1999).

## 3. Results

### 3.1. Macroscopic Characteristics of Pig Kidneys

Pig kidneys were collected at a slaughterhouse in Irapuato, Guanajuato, Mexico. At the first sanitary inspection all pigs were apparently healthy. However, after slaughter some kidneys showed a yellowish coloration. For this reason, a total of 24 kidneys were collected from different pigs. Intentionally, 12 kidneys with yellowish coloration and 12 kidneys with normal (red) coloration were searched, selected and sagittally dissected. Thus, kidneys were classified according to their macroscopic characteristics into reddish kidneys (RK) without apparent damage (Figure 1A), and yellowish kidneys (YK) with possible damage (Figure 1B).

### 3.2. Inflammatory Cell Infiltration in Pig Kidney Tissues

To determine whether the yellowish kidneys were from a pig with a possible infection or inflammatory process, a kidney slice from the renal cortex was evaluated histologically by hematoxylin and eosin staining. Interestingly, cellular infiltrate and inflammatory cells were observed in all collected kidneys, despite macroscopic differences (Figure 2). However, the number of cells was lower in the cellular infiltrate in RK (Figure 2A,B), compared to YK (Figure 2C,D), which was verified by counting the infiltrating inflammatory cells in some RK and YK samples (Figure 3).

### 3.3. Detection of the 16S Subunit of the Bacterial Ribosome and E. coli Strain Identification

During the processing and storage of pork, it has been found that the food products are often contaminated with different bacterial strains. Thus, identification of the 16S rRNA gene was performed by conventional PCR. Unexpectedly, the 16S subunit was detected in all kidneys regardless of macroscopic differences. Figure 4 shows PCR amplification of the 16S ribosomal subunit in six different samples in RK (RK1–6) and six different samples in YK (YK1–6) (Figure 4A,B, respectively).

Because *E. coli* strains have commonly been found to be associated with illnesses caused by ingestion of contaminated food, specific primers for *E. coli* were used to determine the presence of these bacteria in pig kidneys. Surprisingly, Figure 4C,D show that *E. coli* was detected in both RK and YK samples. However, in some YK samples, the amplified product was smaller than in RK samples.

A DNA sample from the small intestinal microbiota of C57BL/6 mice was used as a positive control for both 16S ribosomal subunit detection and *E. coli* identification.

### 3.4. CNF-1 Toxin Detection

Regardless of the macroscopic characteristics of the pig kidneys, both *E. coli* strains and leukocyte cell infiltration were detected in all samples included. However, because YK showed apparent damage, detection of the CNF-1 toxin was performed by conventional PCR. The CNF-1 toxin is expressed by pathogenic bacteria such as *E. coli*, including UPEC strains which can invade the genitourinary tract and colonize both the bladder and kidney. Unexpectedly, amplification of the *cnf-1* gene fragment was found in all kidney samples analyzed (Figure 5). Although it is important to note that dim bands of the amplified products were detected in some RK and YK samples. The amplification of 16S ribosomal subunit was used as an internal control.

In addition, the protein expression levels of the CNF-1 toxin were determined by Western blotting. Consistently, Figure 6 shows that the CNF-1 toxin was detected in all kidney samples analyzed. However, it is important to note that CNF-1 toxin protein levels were higher in YK compared to RK. It is important to note that with the antibodies used in this assay, the 2 bands observed correspond to 110- and 115-kDa detecting CNF1 and CNF2, respectively [27]. Expression of β actin was used as an internal control.

## 4. Discussion

To date, the food industry faces the important challenge of reducing the risks of consuming pork meat infected or contaminated by bacteria, as well as its derivatives such as sausages. Different parts of the pig, such as fat, blood and viscera, such as kidneys, are usually used for the preparation of these products, representing a significant risk to human health [28,29,30]. In this study, *E. coli* was detected by conventional PCR and the CNF-1 toxin was also separately detected using both conventional PCR and Western blot assay in kidneys of apparently healthy pigs from a slaughterhouse in Irapuato, Guanajuato, Mexico, suggesting that the consumption of this pork, as well as processing of products from the viscera of these infected pigs, may represent a significant health risk to humans.

Here, our first findings were made by direct observation in the slaughterhouse, detecting coloration changes and macroscopic differences in kidneys of different pigs. We found that some pig kidneys had a normal reddish coloration (with no apparent damage), while others had a yellowish coloration (with visible damage like hydronephrosis). Renal disorders such as polycystic kidney disease (PKD), a hereditary disorder [31], hydronephrosis [32] and pyelonephritis [33,34] are commonly found in pigs. Interestingly, infection with pyelonephritis-inducing *E. coli* has been detected in slaughter pigs and sows [33]. Likewise, genetic homology has been detected between bacteria infecting both food-producing animals and humans, particularly in *E. coli* [35]. In addition, it is important to mention that meat and meat products can be contaminated at different stages of the food chain, from the slaughterhouse during evisceration to the processing stage, as well as cross-contamination of the food environment and failure to cook meat properly [36,37,38].

It is well known that the consumption of meat by humans is a potential route of transmission of antimicrobial resistance, and food-producing animals have been associated as a major reservoir of resistant bacterial strains, including *E. coli* [38,39]. Unexpectedly, this study found cellular infiltrate and inflammatory cells in all collected kidneys, regardless of macroscopic differences. This suggests that the presence of the inflammatory cell infiltrate in the kidneys was induced by some pathogenic microorganism. Inflammatory cellular infiltrate has been detected in both hydronephrosis and porcine PDK [40,41], which is mainly induced by bacterial infections such as: *E. coli* and *Actinobaculum suis* [42,43]. *E. coli* strains can be classified as pathogenic or commensal and are an important indicator of fecal contamination in food and the food industry. They have also been used as sentinel bacteria to assess antimicrobial resistance in animals and humans. *E. coli* strains can easily survive and spread in diverse environments, so they can be isolated from a variety of sources, such as feces, manure, water and foods of animal and plant origin [44,45].

As mentioned above, *E. coli* has been associated with hydronephrosis and increased inflammatory cell infiltrate in pigs. Herein, we observed yellowish kidneys with apparent damage and a significant leukocyte infiltrate. We decided to determine the presence of *E. coli* bacteria. Surprisingly, *E. coli* was detected by conventional PCR assay in all kidney samples collected. Although specific characterization of the *E. coli* strain was not performed, the CNF-1 toxin was detected by both conventional PCR and Western blot assay. CNF-1 is a cyclomodulin commonly expressed by UPEC and importantly implicated in renal colonization [46]. In addition, UPEC strains have been associated with cystitis and pyelonephritis [47], and different mechanisms have been related to their invasion, survival, bacterial biofilm formation and inflammation [48,49]. Moreover, other virulence factors associated with UPEC have been identified in food animals such as pigs and poultry [50], including Hemolysin A, which induces kidney damage and apoptosis [51,52,53]. However, the invasiveness of UPEC strains has been mainly associated with the expression of the CNF-1 toxin [48,54], which is a potent stimulator of the immune system inducing cytokine production, increasing inflammation and tissue damage [55,56,57].

Herein, the presence of *E. coli* and CNF-1 toxin was determined only in kidneys of pigs from a slaughterhouse, but we cannot discard a general infection in these pigs and that *E. coli* and CNF-1 toxin were present in other organs. Thus, direct consumption of undercooked pork or food products inadequately processed for human consumption could be a possible reservoir of pathogenic *E. coli* strains. In this regard, it has been widely described that the Shiga toxin-producing enterohemorrhagic *Escherichia coli* (STEC) strain is mainly transmitted by poultry, cattle and their meat products [58]. The digestive tract of ruminants has been considered the main reservoir of STEC. Therefore, undercooked beef and unpasteurized milk have been considered high-risk foods for STEC infections [38]. For this reason, the food industry has used various physical and chemical methods to control *E. coli* O157:H7 contamination [59,60]. Although these methods are highly effective, they are not applicable to all foods because they can alter sensory properties, such as color, flavor, texture and general appearance, or produce toxicity [61,62]; which has a significant effect on consumer acceptance. Therefore, alternative methods with biocontrol agents have been developed in recent years [45]. Interestingly, reports on the detection of CNF-1-producing *E. coli* in naturally infected pigs used in the food industry do not exist. However, extra-intestinal pathogenic *E. coli* (ExPEC), including UPEC, has been detected in poultry and swine slaughterhouses in Germany [63]. Although CNF-1 has not been identified in pig kidneys, the toxins have been found in weaned pigs [64] and also in carcasses and minced meat of pork, lamb and beef [65]. In addition, CNF1-producing *E. coli* strains have been detected in various organs of experimentally infected neonatal pigs [66].

Our findings consistently demonstrated the presence of the CNF-1 toxin in all pig kidneys analyzed. However, the results showed differences in gene amplification and protein expression levels of CNF-1 toxin, such as in the YK3 sample. Since CNF-1 expression is regulated by norepinephrine, a hormone produced by the adrenal glands; it is possible that the presence of the CNF-1 producing UPEC strains alters renal function. Thus, it has been previously shown that low levels of norepinephrine increase CNF-1 toxin expression, whereas high levels interfere with CNF-1 expression [67]. Additionally, the differences in *E. coli* detection and CNF-1 levels may be due to pathogenicity and virulence of the *E. coli* strains [66].

On the other hand, it is well known that antimicrobial-resistant bacterial infections are more prevalent in low- and middle-income countries than in developed countries. Worldwide, foodborne infections are a leading cause of morbidity and mortality in humans. Zoonotic microbes can spread through different pathways, including the environment, animals, humans and the food chain. Thus, although antimicrobial drugs are used to treat infections in humans and animals, as well as prophylactically in production agriculture, it is common that food can be contaminated with pathogenic microorganisms from the farm, slaughterhouse and processing for the consumer [68]. Therefore, it is very important that multidisciplinary approaches should be implemented globally to control the spread of foodborne pathogens and promote food safety and security.

It is very important to underline that we did not identify the source of contamination of pig kidneys from the slaughterhouse. However, we can suggest that (1) the presence of *E. coli* and the CNF-1 toxin in pig kidneys is the result of the constant use of antibiotics on pig farms, which has led to bacterial resistance to antibiotics, (2) contamination during slaughtering and processing techniques of pork at the slaughterhouse, and (3) improper handling and maintenance of pigs on farms through contamination of feed and drinking water.

## 5. Conclusions

In conclusion, in this study we demonstrated contamination by CNF-1 toxin-producing *E. coli* in pig kidneys in a slaughterhouse, even in those without apparent damage. This suggests that pork may serve as a reservoir for pathogens and that direct consumption of undercooked meat or processed products from infected pigs and their consumption could be a major human health concern.

## Figures and Tables

**Figure 1 microorganisms-11-02065-f001:**
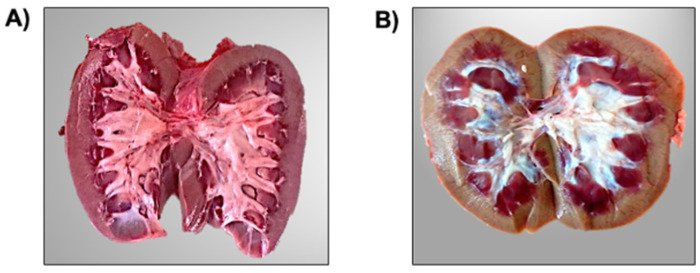
Macroscopic Characteristics of Pig Kidneys. Pig kidneys were sagittally dissected. Reddish kidneys (RK) without apparent damage and yellowish kidneys with possible damage (YK) are shown in (**A**,**B**), respectively.

**Figure 2 microorganisms-11-02065-f002:**
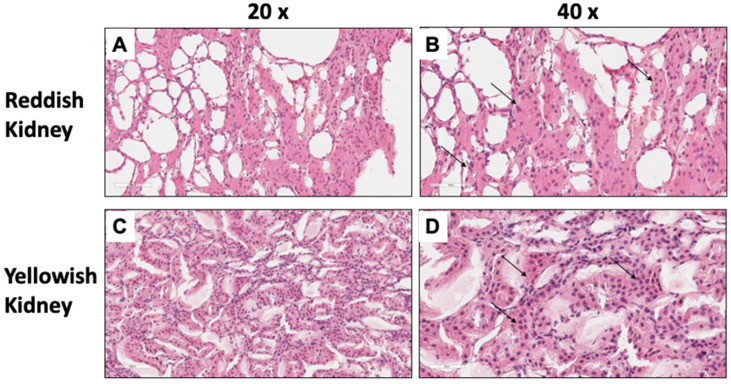
Cellular infiltration in pig kidney tissues. A slice of renal cortex was evaluated histologically by hematoxylin and eosin staining and analyzed by conventional light microscopy at 20× and 40× magnification, respectively. Cellular infiltrate was observed in both reddish kidney (**A**,**B**) and yellowish kidney tissue (**C**,**D**). The arrows indicate the cellular infiltrate in each sample.

**Figure 3 microorganisms-11-02065-f003:**
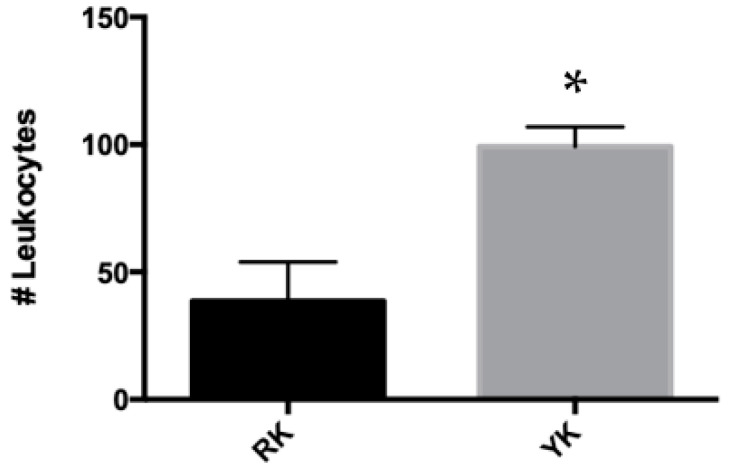
Quantitative assessment of inflammatory cell infiltrate. Cellular infiltration was analyzed by conventional light microscopy, counting 10 different fields of each kidney sample. The bars represent the average of the infiltrating inflammatory cell counts in at least 5 different RK (black bar) and YK (gray bar) samples, respectively. * *p* < 0.05 by U-Mann–Whitney test.

**Figure 4 microorganisms-11-02065-f004:**
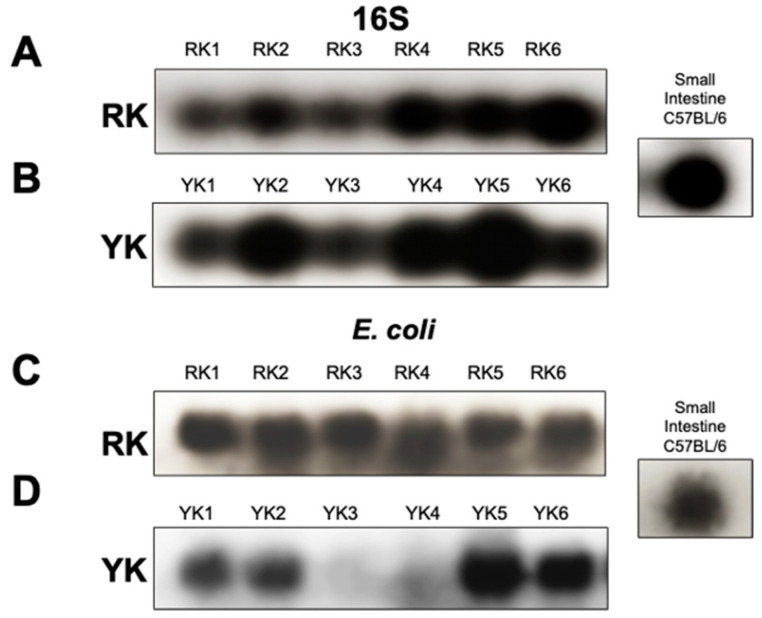
Detection of bacterial ribosome 16S subunit and *E. coli* identification. Total DNA was purified from renal cortex tissue. Amplification of 16S ribosomal subunit and *E. coli* specific gene was performed by end-point PCR. Amplification of the 16S ribosome subunit (**A**,**B**) and the *E. coli*-specific gene (**C**,**D**) is shown in six different samples of reddish kidneys (RK 1–6) and six different samples of yellowish kidneys (YK1–6), respectively. As a positive control for both 16S ribosome subunit and *E. coli*, a mouse intestinal microbiota DNA sample was used.

**Figure 5 microorganisms-11-02065-f005:**
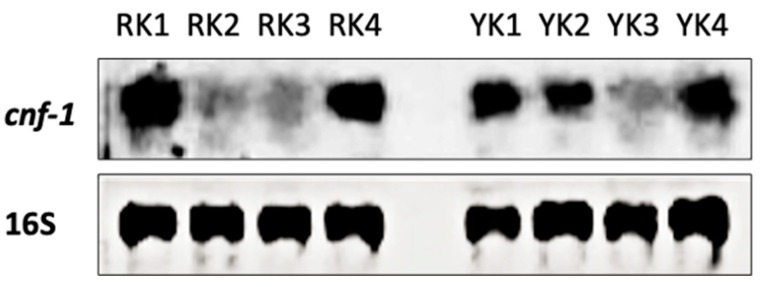
Gene amplification of *cnf-1*. Total DNA was purified from renal cortex tissue. Gene amplification of *cnf-1* was performed by endpoint PCR. The *cnf-1* gen is shown in four different samples of reddish kidneys (RK 1–4) and four different samples of yellowish kidneys (YK1–4), respectively. Amplification of the 16S ribosomal subunit was used as a control.

**Figure 6 microorganisms-11-02065-f006:**
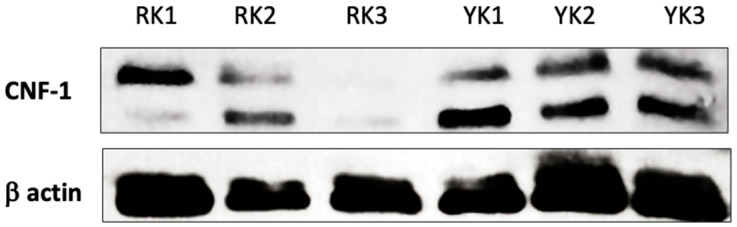
Detection of CNF-1 protein expression levels. Total protein was purified from renal cortex and subjected to SDS-PAGE. CNF-1 toxin protein was determined by Western blot assay. The protein expression levels of CNF-1 toxin are shown in three different reddish kidney samples (RK 1–3) and three different yellowish kidney samples (YK1–3), respectively. Detection of β-actin was used as a control to compare CNF-1 toxin expression levels.

## Data Availability

Data available: The data presented in this study are available on request from the corresponding author.

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
