# Peer review of "Detection of Cyclomodulin CNF-1 Toxin-Producing Strains of Escherichia coli in Pig Kidneys at a Slaughterhouse"

_microorganisms, 2023, doi:10.3390/microorganisms11082065_

Round 1

Reviewer 1 Report

The manuscript presents a rather interesting aspect of the potential contamination of food of animal origin with uropathogenic strains of E coli.

However, a few points need clarification:

It is not entirely clear what the division of kidneys by colors means: as I understand it, YK is a pathologically changed kidney, while RK means a kidney taken from a healthy animal without clinical symptoms?

It would be better to describe the nature of the toxin, first of all, its action and whether it is thermostable or thermolabile, which is important in the technological processing of food

It is puzzling that the toxin was detected in all samples. What safety rules did the authors use when collecting samples to prevent the contamination process. It is worth describing this process in Materials and methods.

It would be appropriate to discuss the YK3 sample, which in all three figures (4,5,6) gives a dubious result

English needs a little correction in a few places regarding vocabulary, e.g. line 72, line 139

Author Response

Reviewer 1
The manuscript presents a rather interesting aspect of the potential contamination of food of animal origin with uropathogenic strains of E coli.
However, a few points need clarification:
It is not entirely clear what the division of kidneys by colors means: as I understand it, YK is a pathologically changed kidney, while RK means a kidney taken from a healthy animal without clinical symptoms?
RESPONSE: In the material and methods section The classification of kidneys that we have used in this study has been better described. We hope it is sufficiently clear. (Lines 80-91)
It would be better to describe the nature of the toxin, first of all, its action and whether it is thermostable or thermolabile, which is important in the technological processing of food
RESPONSE: In the introduction section, the functions of the toxin have been briefly described. (Lines 71-74)
Regarding to the temperature stability of toxin, we did not found reports indicating whether it is thermostable or thermolabile. Thus, because it is a protein; we assume that it is sensitive to denaturation by temperatures used in cooking and food processing. Therefore, in the discussion section, it is mentioned that, "direct consumption of undercooked pork or food products
inadequately processed for human consumption could be a possible reservoir of pathogenic E. coli strains. (lines 300-303)
It is puzzling that the toxin was detected in all samples. What safety rules did the authors use when collecting samples to prevent the contamination process. It is worth describing this process in Materials and methods.

RESPONSE: It is certainly disconcerting and also worrying that we detected the toxin in all samples. In the material and methods section, we have described the biosafety measures we used in handling and transferring the samples to our laboratory. We hope this information is sufficient. (lines 80-91)
It would be appropriate to discuss the YK3 sample, which in all three figures (4,5,6) gives a dubious result
RESPONSE: A brief explanation of the possible mechanisms involved in the differences in E. coli detection and CNF-1 levels is provided in the discussion section. (Lines 321-328)
Comments on the Quality of English Language
English needs a little correction in a few places regarding vocabulary, e.g. line 72, line 139
RESPONSE: these have been corrected

Reviewer 2 Report

In this manuscript the authors detect E. coli and CNF (gene and product) in pig kidneys.

Main comments are:

the authors mix up detection of E. coli and CNF with detecting UPEC involved CNF producing E. coli. There is no direct link between the E. coli and the CNF detection. Although this can be accepted, as E. coli is the known carrier of the cnf gene. Still it needs a comment and discussion, as the authors separately detect the two, and never together. They could for instance have isolated E. coli and tested these for cnf. More important is the mix up with UPEC. The authors indicate at several places in the manuscript that cnf is produced by pathogenic E. coli including UPEC. But in other places in the manuscript they use the detection of cnf as prove that they have found UPEC in pig kidneys. that is not correct, and needs to be changed. For instance in line 263.

The other major comment is that the discussion is more a summary of results and review of literature. It should be a discussion of the results. This needs a lot of adaptation. Discuss the results and compare to other studies. It is now unclear if other people also find E. coli or CNF or cnf producing E. coli in pig kidneys, or other animals. And how many, what techniques. And discuss technicalities, such as why are there two bands in CNF detection?

More minor:

E. coli should be in italics (except when the title of a paragraph is all in italics). In the title Coli should be without capital. 

Indication of strain should often be strains. You are not trying to detect one specific strain but strains that are producing CNF. For instance in title and abstract.

1st paragraph of the Intro has two ways of writing foodborne.

E. coli detection gene should be indicated, with reference to where primers are coming from.

reference to cnf primers is secundary. the paper it refers to, refers to another paper. And why is the first A of the primer missing for the fwd primer?

Line 154 should be in the discussion.

line 170 can be deleted. 16S does not need explanation

Why is figuren 5 so bad. It does not need to be so overexposed. better to have less exposure.

Why do we go from 6 samples analysed, to 4, to 3. this needs explanation and discussion.

Discussion now says sausages are made of kidneys. Should be "can be part of" or something similar.

Line 241: what's PDK?

Reference 15, title is incomplete.

Some untidiness, that needs to be checked. Strain when it should be strains. Line 108 Capital For should be removed. Line 208 extra space. Line 54 should be bacterial strains.

Nothing big, just needs a good read through. 

Author Response

Reviewer 2
Main comments are:
the authors mix up detection of E. coli and CNF with detecting UPEC involved CNF producing E. coli. There is no direct link between the E. coli and the CNF detection. Although this can be accepted, as E. coli is the known carrier of the cnf gene. Still, it needs a comment and discussion, as the authors separately detect the two, and never together.
They could for instance have isolated E. coli and tested these for cnf. More important is the mix up with UPEC. The authors indicate at several places in the manuscript that cnf is produced by pathogenic E. coli including UPEC. But in other places in the manuscript they use the detection of cnf as prove that they have found UPEC in pig kidneys. that is not correct, and needs to be changed. For instance, in line 263. This has been corrected.
RESPONSE: Thank you very much for the comment and clarification, the information has been corrected throughout the manuscript, hopefully it will be less confusing (highlighted in yellow)

The other major comment is that the discussion is more a summary of results and review of literature. It should be a discussion of the results. This needs a lot of adaptation. Discuss the results and compare to other studies. It is now unclear if other people also find E. coli or CNF or cnf producing E. coli in pig kidneys, or other animals. And how many, what techniques.
RESPONSE: Thank you for your comments. We have tried to improve the discussion. However, the little information that exists on cnf-producing E. coli in pig kidneys, or other animals does not facilitate the comparison of our findings with the results obtained by other researchers. Nevertheless, we have incorporated some information and references in an attempt to better support our results (lines 312-320)

And discuss technicalities, such as why are there two bands in CNF detection?
RESPONSE: In the results section, some information has been added that the antibodies used in this assay, the 2 bands observed correspond to 110- and 115-kDa detecting CNF1 and CNF2 respectively. (Lines 236- 238)

More minor:
E. coli should be in italics (except when the title of a paragraph is all in italics).
RESPONSE: This has been corrected throughout the manuscript

In the title Coli should be without capital.
RESPONSE: This has been corrected

Indication of strain should often be strains. You are not trying to detect one specific strain but strains that are producing CNF. For instance in title and abstract.
RESPONSE: This has been corrected throughout the manuscript

1st paragraph of the Intro has two ways of writing foodborne.
RESPONSE: This has been corrected

E. coli detection gene should be indicated, with reference to where primers are coming from. reference to cnf primers is secundary. the paper it refers to, refers to another paper. And why is the first A of the primer missing for the fwd primer?
RESPONSE: the correct reference (26) has been included

Line 154 should be in the discussion.
RESPONSE: This line has been removed from the results and moved to the discussion section.

line 170 can be deleted. 16S does not need explanation
RESPONSE: This has been removed

Why is figure 5 so bad. It does not need to be so overexposed. better to have less exposure.
RESPONSE: Thank you very much for the comment and critique. We believe that despite the overexposure of the gel shown in the figure 5, it shows sufficient quality to demonstrate the amplification of cnf-1 gene performed by end-point PCR. Therefore, we hope that this figure, in spite of its quality, is sufficient to demonstrate the above mentioned.

Why do we go from 6 samples analyzed, to 4, to 3. this needs explanation and discussion.
RESPONSE: The difference in the number of samples shown in the results section is purely methodological. That is, depending on the work capacity of each person who performed the experiments was the number of samples shown in each result obtained. However, it is worth mentioning that all 24 kidney samples were analyzed, but only one representative result from each experiment is shown. Therefore, we respectfully believe that it is not necessary to include an explanation or discussion. Since, as mentioned above, this is a purely methodological maneuver used by those in charge of conducting the experiments.

Discussion now says sausages are made of kidneys. Should be "can be part of" or something similar.
RESPONSE: This has been corrected (lines 250-251)

Line 241: what's PDK?
RESPONSE: The meaning of PDK has been described. (Line 262)

Reference 15, title is incomplete.
RESPONSE: This has been corrected (line 398)

Round 2

Reviewer 1 Report

I have no any other comments